# Evaluation of the Environmental Benefits Associated with the Addition of Olive Pomace in the Manufacture of Lightweight Aggregates

**DOI:** 10.3390/ma13102351

**Published:** 2020-05-20

**Authors:** Manuel Uceda-Rodríguez, Ana B. López-García, José Manuel Moreno-Maroto, Carlos Javier Cobo-Ceacero, María Teresa Cotes-Palomino, Carmen Martínez García

**Affiliations:** Department of Chemical, Environmental and Materials Engineering, Higher Polytechnic School of Linares, University of Jaen, Scientific and Technological Campus of Linares, 23700 Linares (Jaén), Spain; muceda@ujaen.es (M.U.-R.); ablopez@ujaen.es (A.B.L.-G.); jmmaroto@ujaen.es (J.M.M.-M.); cjcobo@ujaen.es (C.J.C.-C.); mtcotes@ujaen.es (M.T.C.-P.)

**Keywords:** life cycle assessment, alperujo, olive pomace, circular economy, lightweight aggregate, waste recycling

## Abstract

A Life Cycle Assessment (LCA) using SimaPro software has been carried out concerning the manufacture of artificial lightweight aggregates (LWAs). The study aims to evaluate the changes in the environmental impact when an additive of residual origin, specifically olive pomace (OP), is added following the principles of the Circular Economy. This residue (commonly known as alperujo) was used as a substitute for clay in 1.25, 2.5 and 5 wt%. The environmental impact related to the use of olive pomace in the mixture was estimated using the CML 2000 methodology, yielding improvements of 3.8%, 7.7% and 15.3% for 1.25, 2.5 and 5 wt% OP added, respectively. Optimum addition results are in the range of 1.25 and 2.5 wt% OP. In this way, the reduction of emissions associated with LWA manufacture would be favored without negatively affecting the technological properties of the resulting material.

## 1. Introduction

The consumerist lifestyle and the linear production systems carried out by industries have meant that society is witnessing an exponential increase in the generation of by-products and waste whose main destination is landfill [1]. Furthermore, in response to the evident global warming, it is essential to carry out productive systems based on the reduction and efficient use of energy. This has contributed to the adoption by countries and organizations of a large number of regulations and standards aimed at the Circular Economy, achieving greater global awareness, a necessary aspect for the preservation of the environment [2]. 

The construction sector is one of the least environmentally sustainable, generating high environmental costs, mainly due to the high consumption of resources and the large amount of waste produced. Aggregate is the second most consumed raw material by man, only behind water. Its main destination is the construction sector, followed by industry and environmental protection, which gives it a clear strategic character [3]. The strong environmental impact that this implies, raises the manufacture of artificial aggregates as a possible alternative to natural aggregates. The manufacture of artificial aggregates allows for a more thorough control process of the raw material, which can lead to materials with specific properties that may even exceed those of traditional aggregates. In particular, due to its low density, high porosity, inert character and reasonable mechanical strength, artificial lightweight aggregate (LWA) is suitable for a wide variety of applications and its growth in different fields is increasing every day [4]. 

According to Ayati et al. [5], natural clays and clayey residues are currently the most suitable raw materials for LWA production due to their particular characteristics and high availability in urban areas. The growing demand for lightweight concrete and thermal insulation may increase its market in the coming years. In the case of a linear economy, the extraction of the main raw material, clay, would come from the same quarries currently dedicated to the production of ceramic materials for construction. However, this activity generates an environmental impact that would imply the restoration of the affected areas. An even more environmentally friendly character can be achieved by using other types of waste as well (e.g., sewage sludge, residues generated from the metallurgical and mining industry, etc.), an aspect tackled by Dondi et al. [6]. Such valorization would not only allow the elimination of materials previously catalogued as *waste* by giving them a second life, but it could even improve the properties of the final product.

Spain is one of the largest producers of olive oil globally, generating more than 1.2 million tons per year [7]. Specifically, the region of Andalusia, in the south of Spain, has more than 1000 installations dedicated to the transformation of olive into oil [8]. The so-called *alperujo* (olive pomace) is the main waste product of the process. This olive pomace (OP), or wet pomace, is a combination of olive husk and pulp (about 20 wt%), crushed olive stone (about 15 wt%) and residual water from the oil mill, which means a moisture content of about 65% [9,10,11]. According to data provided by AGAPA (2015) [8], out of 5.8 million tons of olives collected annually, more than 4 million tons of pressed waste is generated, representing 65% of the initial weight. If we count the olive stone and other waste, this figure rises to 80%. If we take into account the world production, we would reach 11 million tons of residual olive pomace [12]. The dumping of these wastes in aquatic ecosystems or landfills can cause pollution, generate undesirable odors and increase phytotoxicity [11], so recycling them is really important from an environmental standpoint. In recent years, alperujo has been used as an amendment for agricultural crops [13] as well as in the production of active carbon, hydroxytyrosol and other value-added compounds [14]. The current use of this kind of waste in Andalusia is focused on obtaining energy; 47% is used for electricity generation or cogeneration, and 32.9% for thermal uses. A total of 14.3% is used to incorporate it into the soil as organic matter [8]. 

There is hardly any literature on the application of olive pomace in the construction sector. De la Casa et al. [15] published that the addition of alperujo to the ceramic paste reduces the density and thermal conductivity of the final pieces, thus improving their performance compared to traditional brick, which concurs with the results obtained from other studies [16]. This fact opens up a field with enormous potential for the recovery of this waste, which is generated in large quantities, offering the possibility of its valorization as part of the raw material needed in these production systems.

Life Cycle Assessment (LCA) is one of the most widely used methodologies to evaluate the impact a product has on the environment. This methodology is regulated by ISO 14040 and ISO 14044 standards [17,18], and its main function is to establish a common basis from which to fix the inputs and outputs to the biosphere and technosphere of the product through four steps for each approach: definition of objective and scope, inventory analysis, impact analysis and interpretation of results.

## 2. Methodology 

### 2.1. Life Cycle Assessment of Lightweight Aggregates Containing Olive Pomace

#### 2.1.1. Objective and Scope Definition

The objective of this study is to carry out an LCA to quantify the environmental benefits associated with the manufacture of artificial lightweight aggregates based on clay and the incorporation of an agroindustrial waste product, such as alperujo (Consejo Regulador de la Denominación de Origen Sierra de Segura, La Puerta de Segura (Jaén), Spain ), as an alternative to the final disposal of this waste in a landfill [19]. The LCA carried out in this investigation follow the procedure and order proposed by the ISO 14040 and ISO 14044 standards.

This study is based on a previous experimental study conducted by the authors’ research team [20], in which the characteristics of LWAs incorporating different proportions of this residue are described. Specifically, the percentages of 0, 1.25, 2.5 and 5 wt% of OP have been considered when performing the LCA. The results of this research showed that the addition of OP in low proportions was more positive in obtaining a greater expansion of the aggregates, related to increased pore formation and, therefore, a lighter structure. According to Moreno-Maroto et al. [20], the sintering temperatures that have been used to conduct the LCA correspond to "the maximum allowed by the pellet without melting or sticking to the tube or to other pellets". The sintering times and temperatures used for each sample type are specified in Section 2.1.2.

In this study, the manufacture of 1 kg of artificial lightweight aggregates with an apparent density of between 400 and 1200 kg/m^3^ has been established as a functional unit [21,22]. Subsequently, the manufacture of residue-free aggregates produced only with clay will be compared to those manufactured with the same clay but containing OP as an additive. Variations in weight percentages and sintering temperatures have been taken into account.

The scope of this analysis will take a “cradle-to-door” approach, where the impact of the following steps will be analyzed:Raw materials, which would contain both the clay and the OP residue, as well as the water needed for the process. In this stage, the obtaining of the waste has been considered as a process with no environmental load, since otherwise the impact associated with obtaining it would be added to the manufacture of the LWA.Extraction, which would contain the processes related to the occupation of the land used by the quarry, as well as the energy required for the extraction of the clay.Transport of the raw material to the factory has been determined so that the transfer of the raw material would take place within a radius of 50 km, assimilating the transport of the waste to the corresponding part of the clay it replaces, in cases where it affects.Manufacture of the final product, considering the raw materials needed to complete the final product, the energy consumed for this purpose, and the emissions into the air and water.

Figure 1 shows the different elements involved in each of the stages considered. The service life and end of life stages are excluded (as can be seen with the dotted line in Figure 1 and Figure 2), as their inclusion would not differentiate them from the lightweight aggregates without waste, because these two stages are not susceptible to any alteration with respect to the origin of the aggregates.

#### 2.1.2. Systems Limits and Scenarios

The mixture is made up of Spanish white clay (SW), supplied by the company Comercial Cerámicas de Bailén, S.A. (Bailén, Spain) and by the olive pomace (OP) in different percentages, supplied by Consejo Regulador de la Denominación de Origen Sierra del Segura (La Puerta de Segura, Spain). The preparation of the raw materials and the aggregates is explained in Moreno-Maroto et al. [20]. A total of 11 samples have been studied, which have been grouped according to the percentage of waste added (wt%) to establish the different scenarios. In all the scenarios, some common points have been identified, such as the extraction and transport stages. Below are the singularities that distinguish them from each other:LWA scenario without residue (SW-0OP): These aggregates have been manufactured using only clay as raw material. In the raw material phase, only processes related to the clay and water used have been considered. For the manufacture of 1 kg of aggregate, an initial quantity of clay of 1.1 kg is required. The manufacturing stage would include all processes related to the shaping of the final product, including additional materials such as packaging film or transport pallets, the energy required and the emissions produced. These aggregates were sintered at 1205 °C for 4 min, maintaining the aggregates in the preheating zone for 1 min.LWA scenario with 1.25 wt% OP (SW-1.25OP): This scenario corresponds to the lowest waste content in the aggregate, so the raw material and manufacturing stages would have been altered to modify the data affecting the new variable. These aggregates were sintered at 1195 °C for 4 min, plus 1 min in the preheating zone.LWA scenario with 2.5 wt% OP (SW-2.5OP): This mixture was studied in more depth due to the good performance obtained in the previous work [20]. The temperatures and study times were 1140, 1160 and 1180 °C; for 4, 8 and 16 min, and 1 min of preheating. Therefore, a total of nine types of samples were analyzed. The phases of raw materials and manufacturing have been modified to take into account the new addition values and sintering conditions.LWA scenario with 5 wt% OP (SW-5OP): As in the case of 1.25 wt% OP, the corresponding steps have been modified to adjust to the new addition values. These aggregates were sintered at 1180 °C for 4 min. As in the other scenarios, the dwell time of the aggregates in the preheating zone was 1 min.

#### 2.1.3. Life Cycle Inventory Analysis

The Life Cycle Inventory Analysis is the main stage of the Life Cycle Assessment. It must incorporate reliable data that most closely represent the reality it aims to model. The sources used to build the life cycle inventory have been (i) data measured and empirically processed in the experimental research carried out in the laboratory, such as the proportions of materials used, energy consumption and measurement of gases emitted in the sintering process; (ii) bibliographic data [23,24]; (iii) background data to complement the processes, extracted from reference values of databases such as Ecoinvent v.3.2 (Ecoinvent, Zurich, Switzerland) [25], based on those processes that most closely resemble the objective and scope of this study, with a global scope and sintering temperatures similar to those used in this life cycle study.

The set of these data has resulted in the modelling of the systems included in Table 1.

#### 2.1.4. Life Cycle Impact Assessment Methodology

The calculation methodology selected was CML 2000 in its version 2.05 (Centre for Environmental Studies, Leiden, The Netherlands), a midpoint-oriented method that was first developed in 1992 by the Institute of Environmental Sciences of Leiden University. The choice of this methodology was based on the following reasons: (i) the versatility offered by the impact categories it calculates, making it possible to know, in a single methodology, the areas of protection corresponding to human health, the environment, the artificial surroundings and natural resources; (ii) the characterization is based on global and European average values; (iii) the wide variety of studies and scientific articles that guarantee its correct operation [26,27,28].

The baseline impact categories analyzed in this version of the methodology are as follows: Abiotic Depletion, Acidification, Eutrophication, Global Warning Potential, One-Layer Depletion, Human Toxicity, Fresh Water Aquatic Ecotox, Marine Aquatic Ecotoxicity, Terrestrial Ecotoxicity and Photochemical Oxidation.

With regards to this methodology, the results have also been evaluated from a standardized point of view regarding the environmental effects caused by an average European citizen in one year. This allows us to obtain a more realistic view of the relative importance of the different environmental effects.

The computer software used in this study was SimaPro version 8.3.0.0 from PRé Consultants (Amersfoort, The Netherlands). In Figure 2, we have outlined the elements to form a Life Cycle Inventory (LCI) with which we have simulated the process in SimaPro.

#### 2.1.5. Contribution and Influence Analysis

Given the objective and scope of this work, it has been considered useful to structure and analyze the data obtained according to their contribution and influence. The contribution analysis aims to examine the contribution of the different stages of the life cycle to the total result, expressing this contribution as a total percentage, with the results being classified as follows: significant influence (> 50%), relevant influence (25%–50%), some influence (5%–25%), minor influence (2.5%–5%) or negligible influence (<2.5%) [17]. In turn, this assessment was accompanied by an influence analysis, which examines the possibility of influencing, to a lesser or greater extent, the environmental factors affected. In this way, it was assessed whether there was significant control, with large possible improvements (A), little control, with some possible improvements (B), or, conversely, no possible control (C). The degree of control can be interpreted as the capacity of the production company to modify the current processes in order to reduce the contribution.

## 3. Results and Discussion

### 3.1. Impact Analysis of Lightweight Aggregates Without Residue

The impacts associated with the life cycle of 1 kg of residue-free lightweight aggregates manufactured entirely from clay are shown in Table 2 and Figure 3. The most significant impact of the system is on climate change (Global Warming Potential) with values of 0.37 kg eq. CO_2_/kg LWA, a value in line with the results obtained by other authors [29]. Most of this impact is attributed to the manufacturing stage (88%) due mainly to the combustion of fossil fuels during the sintering process. The manufacturing stage is a very important part of the production process and must therefore be controlled with great precision because the kilns must be working at very high temperatures, above 1000 °C, with the great economic and environmental cost that this entails. The next stage with the greatest contribution is extraction, which represents 12% of CO_2_ emissions. The third stage, in order of contribution, would be the transport which takes a value of 3%. Finally, the raw materials phase would only represent 2% of the total in this impact category.

If we take into account the other impact categories, we can see how the contribution values are similar to those obtained for the Global Warning Potential category. In the case of One-Layer Depletion the contribution of the manufacturing stage decreases to 60%, producing an increase in the rest of the stages: in transport the contribution increases to 19%, while the contributions associated with the extraction and raw material stages rise to 12% and 10%, respectively. In the same way, other impact categories such as Eutrophication, Human Toxicity, Fresh Water Aquatic Ecotox, Marine Aquatic Ecotoxicity and Terrestrial Ecotoxicity show values for the manufacture of 80 % of contribution, and between 12%-17% for the extraction stage.

These very similar values are mainly due to the strong interrelationship between them, due to the emission of greenhouse gases (GHG), with carbon dioxide being the second largest GHG after water vapor. Approximately 50% of CO_2_ emissions are retained in the atmosphere, while the remaining 50% is incorporated into the ocean and the terrestrial biosphere [30], decoupling the natural balance of exchange between these and the atmosphere [31]. In addition to destabilizing the climate, CO_2_ has a serious and strong impact on the oceans. The oceans account for 30% [32] of global emissions and 80% of the heat generated by the growing increase in greenhouse gases due, among other things, to industry and the combustion of fossil fuels by jet engines. An increase in water temperature and atmospheric CO_2_ concentration causes the dynamic balance of ocean pH and water in general to break down, increasing their acidity and impacting on the marine biosphere. In turn, the discharge of waste from the same emitting industries, rich in nitrogen, phosphorus and organic matter to aquatic ecosystems and the coast, causes a problem of eutrophication, increasing the metabolic rate of the organisms, and causing the loss of water and sediment quality. The impacts of global change on ecosystems, which are increased by this type of emissions, eventually affect the physical factors on which human health strongly depends (the biogeophysical environment, food, the genetic base, etc.) Although they may seem like independent aspects, all actions have an impact on the ecosystem to which we all belong and therefore affects us.

With regards to the degree of influence, the manufacturing stage would be associated with a grade A, with significant control and large possible improvements. At this stage there are different points where modifications could be made, such as reducing the dwell time in the kiln or the sintering temperature. Any reduction in this aspect influences energy consumption and, therefore, the environmental impact. Furthermore, the addition of other types of materials, especially organic waste, whose thermal decomposition is exothermic, plays a key role in reducing the energy input required to reach the sintering temperature. The stages of extraction and transport would be associated with a C level, since there is not a sufficiently broad control of action to alter the current procedures. The final stage of raw materials would be associated with a B level since, although it is possible to introduce variations or modifications in the typology or quantities of material, to a large extent these must maintain a relative constancy so that the properties of the resulting aggregates are adequate.

### 3.2. Impact Analysis of Lightweight Aggregates Containing Olive Pomace

The results shown in Table 2 include the data from the aggregates containing the olive pomace residue. These show a similar behavior to that of the residue-free aggregates; however, there are some variations, mostly beneficial from an environmental point of view, which are worth discussing. This is an expected result and would confirm that the study has been carried out correctly. We are talking about minimal differences between the manufacture of the residue-free LWA and those in which OP is used as an additive. This is again reflected in the high percentages of contribution of the manufacturing stage in the different impact categories analyzed, mainly in the emission of GHGs with values exceeding 80%.

Considering an overall assessment of the results between the different impact categories analyzed, the addition of OP in percentages of 1.25, 2.5 and 5 wt% has slightly modified the pattern of results previously shown for the aggregates without additive, as detailed below. In all the impact categories, the raw materials stage represents barely 1%-3% of the contribution in the different mixes studied, so it would have a negligible influence, except in One-Layer Depletion, where the contribution would rise to a value that ranges between 9% and 11% depending on the type of aggregate, with the maximum value for the mixtures containing 5 wt% OP and some mixtures with 2.5 wt% OP. The degree of influence would be B, with little control, and some possible improvements, since we can partially modify the composition of the mixture and alter the quantities of clay and additive.

The second extraction stage undergoes a progressive decrease, now showing a value of 9.4%, 8% and 5%, for 1.25, 2.5 and 5 wt%, respectively, in the Global Warning Potential category, so the latter would now have a contribution with a lower influence than in the residue-free aggregates, but the degree of influence would still be maintained, C. The remaining categories for the 1.25 wt% OP sample, show fluctuations of between 4% for Acidification and Photochemical Oxidation, and a maximum of 16% in Fresh Water Aquatic Ecotox. The other categories would have values between a 6% and 13% contribution. The LWAs manufactured with 2.5 wt% OP present a similar distribution in terms of categories, in this case, related to minimum contribution values of 3% and maximum ones of 14%, with a slight variation on these figures depending on the different time and temperature settings. In the specific case of the mixtures containing 5 wt% OP, the One-Layer Depletion category moves to a second scenario. The largest contribution in this stage is the Human Toxicity category, while the smallest contributions (only 1%) would be for Acidification and Global Warning Potential.

The transport stage has also remained largely unchanged, with contributions ranging from 3% to 4%-5% in most of the impact categories and different mixtures. This would correspond to a lower influence and a degree C in this aspect, with the exception of the One-Layer Depletion category. In this category, the value of the contribution goes up to 18%-22%, being 20% in the mixtures with 1.25 wt% OP, 18%-20% in the different sintering configurations of the mixtures with 2.5 wt% OP, and 22% in the case of the aggregates with 5 wt% OP, which would be a quite important contribution for this stage of the production process.

The last stage increases with each percentage addition, from 89% to 91% for the 1.25 and 2.5 wt% OP, up to 93% for the 5 wt% OP in the Global Warning Potential impact category. This again represents a significant influence, and an A value in the degree of influence, as it is a process that could undergo modifications. The rest of the categories present slightly lower figures, with notable categories such as Acidification, with values between 92%-95% for all the percentages by weight. On the other hand, other categories such as One-Layer Depletion reduce the contribution significantly up to values of 60%, with maximums of 64% in some of the 2.5 wt% OP mixture variants. This would indicate that, although the manufacturing stage has a strong influence, other phases, such as transport, are also really important.

If we compare the results directly with each other, although the differences have not led to noticeable improvements, there have been some decreases in the impact that would benefit the use of the additive in the mixtures. To see the improvement that the addition of OP has supposed, the columns of the results obtained in Figure 4 have been compared with the column of the aggregates without waste, making an average of the values of each impact category. The addition of 1.25 wt% OP has meant a reduction of the environmental impact by 3.8%, while the contribution of 2.5 wt% of additive in the different time and temperature configurations increases this improvement to 7.7% for 4 min of sintering, 1% for 8 min and at 16 min it would exceed the residue-free aggregate by 5.9% on average. The contribution of 5 wt% of additive implies an improvement of 15.3%. Figure 4 shows an example of the reduction of environmental impact for the different categories analyzed from the sample SW-2.5OP sintered at 1180 °C in the different times studied.

### 3.3. Standardized Analysis of Aggregates without Residue and Lightweight Aggregates Containing Olive Pomace

The data from the normalization show results with similar reductions, as can be seen in Table 3 and Figure 5. The graph in Figure 5 shows the mixtures with the different percentages by weight of additives. Only the mixture SW-2.5OP sintered at 1180 °C for 4 min has been included because it has similar sintering characteristics to the rest. It is surprising that the standardized score associated with the marine aquatic ecotoxicity impact category is so far above the other impact categories, even though aggregate production, with or without an additive, is not expected to be a toxic activity for the marine environment. This really makes us wonder whether the values provided by the normalization could be biased in some way, probably because the marine aquatic ecotoxicity is too high or the remaining impact categories are too low. These conclusions are contrasted with several investigations, whose authors came up with results that shared similar trends [23,33,34,35], so it would not be an error in the LCA calculations. The research conducted by Heijungs et al. [36] is noteworthy, in which they describe the phenomenon and its reasons.

Once the data has been analyzed, it has been possible to deduce that the addition of olive pomace reduces the impact related to the production of lightweight aggregates. However, it should be noted that the addition of higher percentages of residues in weight than those shown in this study would generate an increase in emissions with respect to the aggregate without the additive, which would cause an increase in the environmental impact. According to the results, the reduction in greenhouse gas emissions occurs for the samples with a substitution of 1.25 to 3 wt% OP. These results verify those obtained in the previous work of Moreno-Maroto et al. [20], on the production of aggregates with addiction of olive pomace, where it is indicated that the optimum amount of substitution of part of the clay by waste is 2.5 wt% OP, without this negatively affecting the technological properties of the resulting aggregates.

Regarding the aggregates manufactured at different temperatures and sintering times, those in which the firing time was 16 min did not present a substantial improvement at a technological level that justified the greater impact generated.

## 4. Conclusions

The Life Cycle Assessment (LCA) represents a potent tool for evaluating environmental impacts, an increasingly important aspect in any sector. The LCA carried out in this research has examined the environmental impact of the manufacture of lightweight aggregates when different proportions of olive pomace (*alperujo*) are added to clay mixtures, comparing these results with those obtained from the residue-free clay. The data were processed with SimaPro software and using the CML 2000 methodology. The guidelines of the ISO 14040 standard were followed. 

Currently, studies analyzing the life cycle of aggregates focus on comparing natural aggregates with recycled aggregates from construction and demolition waste (CDW). There are major advantages in recycling CDW from an environmental point of view by reducing the impact on eutrophication and ecotoxicity [37]. However, for this to happen optimally, the recycled aggregate has to comply with strict construction technologies which is not always the case, as indicated by Zhang et al. [38]. Another aspect to consider when conducting a life cycle assessment on recycled aggregates is to establish a suitable functional unit for comparison with other studies. Given the heterogeneity and diverse origins that recycled aggregates may have, the appropriate variables must be found to establish the analysis through a method that combines the effect of quality, mass and market price in the allocation procedure of the recycled aggregate [38]. 

The results indicate that the addition of olive pomace in the mixture is beneficial in terms of reducing the environmental impact compared to that generated by the aggregates without the residue. Thus, the environmental impact related to the use of olive pomace, in this case estimated using the CML 2000 methodology, yielded improvements of 3.8%, 7.7% and 15.3% for 1.25, 2.5 and 5 wt% of waste added, respectively, taking into account an overall assessment of the results between the different categories analyzed. However, if we consider the emission-related GWP category, additive proportions close to and above 5 wt% would entail a negative impact compared to the residue-free aggregates, since the combustion of the additive is directly associated with an increase in GHG emissions. Therefore, this study has verified that the optimum proportion of olive pomace to be added is between 1.25 and 2.5 wt%. In this way, the reduction of emissions associated with the aggregate manufacture would be favored up to 7.4%-7.7%, without having a greater negative impact on the technological properties of the resulting material.

The results derived from this study support the hypothesis that the replacement of part of the clay by olive pomace can represent an environmentally friendly method for the management of such waste, which, in turn, could contribute to impulse the production in those territories strongly connected with the ceramic, construction and agri-food industries. To conclude, it has also been demonstrated that tools such as Life Cycle Assessment provide us with an accurate view of the impact that our activities have on the environment, and therefore the LCA application should be enhanced in the future in all sectors.

## Figures and Tables

**Figure 1 materials-13-02351-f001:**
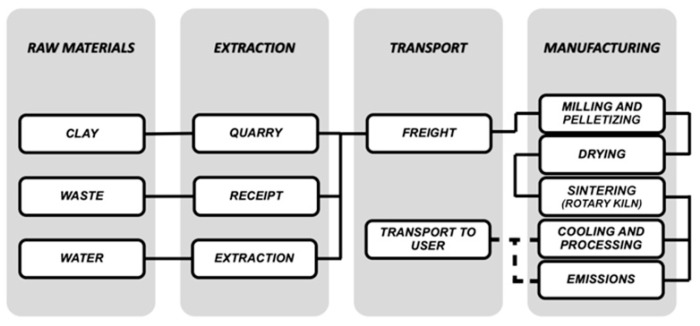
Elements and processes that contemplate each one of the analyzed steps.

**Figure 2 materials-13-02351-f002:**
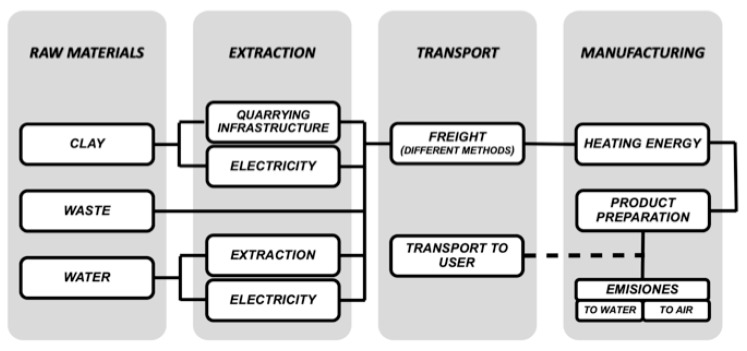
Elements and processes that have been taken into consideration to form the LCI for each step.

**Figure 3 materials-13-02351-f003:**
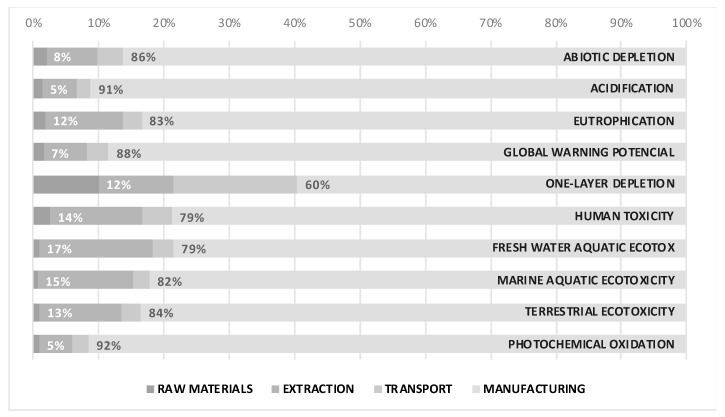
Contribution of characterized impacts associated with 1 kg of lightweight aggregates manufactured without any residue from different stages of production.

**Figure 4 materials-13-02351-f004:**
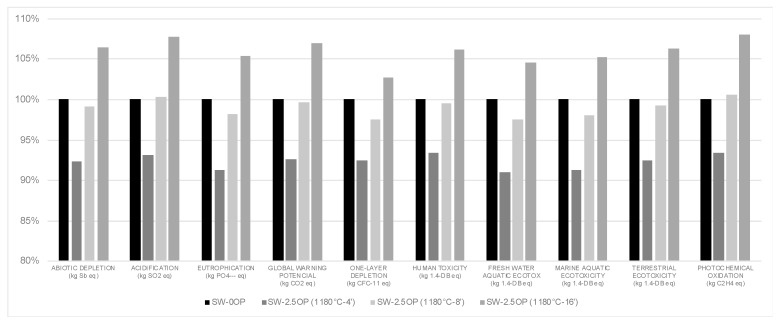
Characterization. Comparison of the impacts obtained between lightweight aggregates produced without waste and those manufactured with 2.5 wt% OP (different sintering conditions).

**Figure 5 materials-13-02351-f005:**
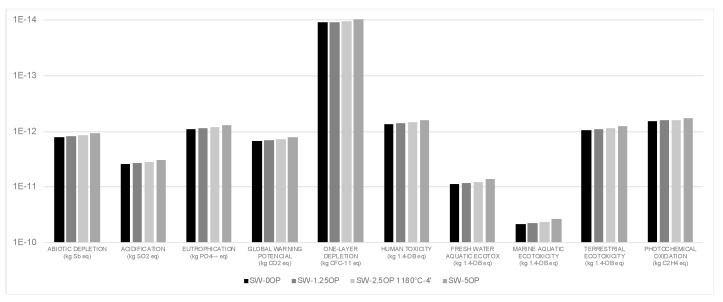
Normalization. Values obtained from different categories for all types of lightweight aggregates studied.

**Table 1 materials-13-02351-t001:** Inventory data adjusted to 1 kg of lightweight aggregates for the different phases analyzed: raw materials, extraction, transport and manufacturing.

Elementary Flow	Units	SW-0OP	SW-1.25OP	SW-2.5OP									SW-5OP	LCIA Dataset
				1140°C-4′	SW-5OP	LCIA Dataset	1160°C-4′	1160°C-8′	1160°C-16′	1180°C-4′	1180°C-8′	1180°C-16′		
Extraction and transport of raw materials														
Clay	kg	1.1	1.08625	1.0725	1.0725	1.0725	1.0725	1.0725	1.0725	1.0725	1.0725	1.0725	1.045	Clay {GLO}| market for | Alloc Def, U
Olive pomace	kg	-	0.01375	0.0275	0.0275	0.0275	0.0275	0.0275	0.0275	0.0275	0.0275	0.0275	0.055	-
Water	m^3^	0.0000736	0.0000736	0.0000736	0.0000736	0.0000736	0.0000736	0.0000736	0.0000736	0.0000736	0.0000736	0.0000736	0.0000736	Water, well, in ground, ES
	kg	0.0134	0.0134	0.0134	0.0134	0.0134	0.0134	0.0134	0.0134	0.0134	0.0134	0.0134	0.0134	Tap water {GLO}| market group for | Alloc Def, U
Transport raw materials	tkm	0.0747	0.0747	0.0747	0.0747	0.0747	0.0747	0.0747	0.0747	0.0747	0.0747	0.0747	0.0747	Transport, freight, lorry, 16-32 tons {GLO}| market for | Alloc Def, U
Extraction plant	p	2.00 × 10^−10^	2.00 × 10^−10^	2.00 × 10^−10^	2.00 × 10^−10^	2.00 × 10^−10^	2.00 × 10^−10^	2.00 × 10^−10^	2.00 × 10^−10^	2.00 × 10^−10^	2.00 × 10^−10^	2.00 × 10^−10^	2.00 × 10^−10^	Clay pit infrastructure {GLO}| market for | Alloc Def, U
Energy and material inputs at LWA manufacturing plant														
Electricity	kWh	0.028881403	0.022153249	0.015425095	0.017592282	0.019882295	0.015423668	0.017590141	0.019880155	0.015421528	0.017588001	0.019878014	0.001957372	Electricity, medium voltage {GLO}| market group for | Alloc Def, U
Heat	MJ	2.549810045	2.482528506	2.415246966	2.606578178	2.808809101	2.415232697	2.606556773	2.808787697	2.415211292	2.606535369	2.808766293	2.280569731	Heat, district or industrial, other than natural gas {GLO}| market group for | Alloc Def, U
Packaging film	kg	0.0004813	0.0004813	0.0004813	0.0004813	0.0004813	0.0004813	0.0004813	0.0004813	0.0004813	0.0004813	0.0004813	0.0004813	Packaging film, low density polyethylene {GLO}| market for | Alloc Def, U
Linerboard	kg	0.002	0.002	0.002	0.002	0.002	0.002	0.002	0.002	0.002	0.002	0.002	0.002	Linerboard {RoW}| market for linerboard | Alloc Def, U
Direct emissions due to thermal transformation of raw materials														
Water	m^3^	0.0000067	0.0000067	0.0000067	0.0000067	0.0000067	0.0000067	0.0000067	0.0000067	0.0000067	0.0000067	0.0000067	0.0000067	Emissions to air - Water/m^3^
CO_2_	m^3^	0.178700361	0.200883398	0.223066435	0.223066435	0.223066435	0.223066435	0.223066435	0.223066435	0.223066435	0.223066435	0.223066435	0.26743251	Emissions to air - CO_2_
Water	m^3^	0.00008568	0.00008568	0.00008568	0.00008568	0.00008568	0.00008568	0.00008568	0.00008568	0.00008568	0.00008568	0.00008568	0.00008568	Emissions to water - Water, RoW

SW-(0-5)OP means Spanish White clay with 0, 1.25, 2.5 and 5 wt% olive pomace addiction; LCIA means life cycle inventory analysis.

**Table 2 materials-13-02351-t002:** Characterized impacts associated with 1 kg of lightweight aggregates incorporating 0, 1.25, 2.5 and 5 wt% OP.

Impact Categories	Units	SW-0OP	SW-1.25OP	SW-2.5OP									SW-5OP
				1140 °C-4′	1140 °C-8′	1140 °C-16′	1160 °C-4′	1160 °C-8′	1160 °C-16′	1180 °C-4′	1180 °C-8′	1180 °C-16′	
Abiotic Depletion	kg Sb eq	0.002161234	0.002077781	0.001994337	0.002142637	0.002299367	0.00199432	0.002142597	0.002299337	0.001994299	0.002142577	0.002299317	0.001827312
Acidification	kg SO_2_ eq	0.002594284	0.002505764	0.002417241	0.002601725	0.002796714	0.002417226	0.002601697	0.002796684	0.002417198	0.002601669	0.002796664	0.00224004
Eutrophication	kg PO_4_^3-^ eq	0.000465992	0.000445684	0.000425377	0.000457324	0.000491096	0.000425375	0.000457321	0.000491093	0.000425371	0.000457318	0.00049108	0.000384734
Global Warning Potential	kg CO_2_ eq	0.37264414	0.35890716	0.34517018	0.37125078	0.39881666	0.34516729	0.37124644	0.39881232	0.34516293	0.37124209	0.39880797	0.31767305
One-Layer Depletion	kg CFC-11 eq	1.11172 × 10^−8^	1.07023 × 10^−8^	1.02875 × 10^−8^	1.08366 × 10^−8^	1.14169 × 10^−8^	1.02875 × 10^−8^	1.08365 × 10^−8^	1.14168 × 10^−8^	1.02873 × 10^−8^	1.08363 × 10^−8^	1.14166 × 10^−8^	9.45718 × 10^−8^
Human Toxicity	kg 1.4-DB eq	0.13830491	0.13372685	0.12914876	0.13772516	0.14678997	0.1291478	0.13772372	0.14678853	0.12914637	0.13772227	0.14678709	0.11998492
Fresh Water Aquatic Ecotox	kg 1.4-DB eq	0.0667779	0.06375513	0.06073238	0.06514047	0.06979952	0.06073174	0.06513951	0.06979857	0.06073078	0.06513855	0.06979761	0.05468174
Marine Aquatic Ecotoxicity	kg 1.4-DB eq	145.8228294	139.47	133.1171707	143.0092696	153.4645963	133.1158261	143.0072527	153.4625794	133.1138082	143.0052358	153.4605625	120.4007532
Terrestrial Ecotoxicity	kg 1.4-DB eq	0.000862918	0.000830666	0.000798401	0.000856102	0.000917088	0.000798398	0.000856098	0.000917084	0.000798384	0.000856084	0.00091707	0.000733829
Photochemical Oxidation	kg C_2_H_4_ eq	0.000121395	0.000117406	0.000113417	0.00012204	0.000131142	0.000113416	0.000122039	0.000131142	0.000113416	0.000122029	0.000131141	0.000105438

SW-(0-5)OP means Spanish White clay with 0, 1.25, 2.5 and 5 wt% olive pomace addiction; (1140–1180)°C indicates the maximum sintering temperature; and, (4-16)′ indicates the time spent in the rotary furnace in min.

**Table 3 materials-13-02351-t003:** Normalized impacts associated with 1 kg of lightweight aggregates incorporating 0, 1.25, 2.5 and 5 wt% OP.

Impact Categories	Units	SW-0OP	SW-1.25OP	SW-2.5OP									SW-5OP
				1140 °C-4′	1140 °C-8′	1140 °C-16′	1160 °C-4′	1160 °C-8′	1160 °C-16′	1180 °C-4′	1180 °C-8′	1180 °C-16′	
Abiotic Depletion	kg Sb eq	1.26436 × 10^−12^	1.21555 × 10^−12^	1.16664 × 10^−12^	1.25348 × 10^−12^	1.34509 × 10^−12^	1.16664 × 10^−12^	1.25347 × 10^−12^	1.34508 × 10^−12^	1.16663 × 10^−12^	1.25337 × 10^−12^	1.34507 × 10^−12^	1.06898 × 10^−12^
Acidification	kg SO_2_ eq	3.8655 × 10^−12^	3.73354 × 10^−12^	3.60169 × 10^−12^	3.87654 × 10^−12^	4.1671 × 10^−12^	3.60168 × 10^−12^	3.87653 × 10^−12^	4.16709 × 10^−12^	3.60157 × 10^−12^	3.87651 × 10^−12^	4.16708 × 10^−12^	3.33762 × 10^−12^
Eutrophication	kg PO_4_^3-^ eq	9.2733 × 10^−13^	8.8692 × 10^−13^	8.46507 × 10^−13^	9.10079 × 10^−13^	9.77278 × 10^−13^	8.46503 × 10^−13^	9.10072 × 10^−13^	9.77262 × 10^−13^	8.46486 × 10^−13^	9.10056 × 10^−13^	9.77255 × 10^−13^	7.65627 × 10^−13^
Global Warning Potential	kg CO_2_ eq	1.47568 × 10^−12^	1.42123 × 10^−12^	1.36687 × 10^−12^	1.47013 × 10^−12^	1.57936 × 10^−12^	1.36687 × 10^−12^	1.47013 × 10^−12^	1.57925 × 10^−12^	1.36686 × 10^−12^	1.47012 × 10^−12^	1.57925 × 10^−12^	1.25794 × 10^−12^
One-Layer Depletion	kg CFC-11 eq	1.13395 × 10^−14^	1.09163 × 10^−14^	1.04932 × 10^−14^	1.10532 × 10^−14^	1.16451 × 10^−14^	1.04931 × 10^−14^	1.10531 × 10^−14^	1.1645 × 10^−14^	1.0493 × 10^−14^	1.10529 × 10^−14^	1.16449 × 10^−14^	9.64632 × 10^−15^
Human Toxicity	kg 1.4-DB eq	7.35789 × 10^−13^	7.11423 × 10^−13^	6.8707 × 10^−13^	7.32694 × 10^−13^	7.80924 × 10^−13^	6.87068 × 10^−13^	7.32691 × 10^−13^	7.80911 × 10^−13^	6.87055 × 10^−13^	7.32688 × 10^−13^	7.80908 × 10^−13^	6.3832 × 10^−13^
Fresh Water Aquatic Ecotox	kg 1.4-DB eq	8.88149 × 10^−12^	8.47943 × 10^−12^	8.07746 × 10^−12^	8.66366 × 10^−12^	9.28336 × 10^−12^	8.07726 × 10^−12^	8.66356 × 10^−12^	9.28316 × 10^−12^	8.07726 × 10^−12^	8.66336 × 10^−12^	9.28306 × 10^−12^	7.27265 × 10^−12^
Marine Aquatic Ecotoxicity	kg 1.4-DB eq	4.57879 × 10^−11^	4.37936 × 10^−11^	4.17983 × 10^−11^	4.49048 × 10^−11^	4.81877 × 10^−11^	4.17981 × 10^−11^	4.49044 × 10^−11^	4.81874 × 10^−11^	4.17977 × 10^−11^	4.49031 × 10^−11^	4.81861 × 10^−11^	3.78059 × 10^−11^
Terrestrial Ecotoxicity	kg 1.4-DB eq	9.40584 × 10^−13^	9.05425 × 10^−13^	8.70263 × 10^−13^	9.33153 × 10^−13^	9.99628 × 10^−13^	8.7025 × 10^−13^	9.33138 × 10^−13^	9.99614 × 10^−13^	8.70245 × 10^−13^	9.33133 × 10^−13^	9.99609 × 10^−13^	7.99871 × 10^−13^
Photochemical Oxidation	kg C_2_H_4_ eq	6.66466 × 10^−13^	6.44573 × 10^−13^	6.2268 × 10^−13^	6.69973 × 10^−13^	7.19962 × 10^−13^	6.22679 × 10^−13^	6.69971 × 10^−13^	7.19961 × 10^−13^	6.22667 × 10^−13^	6.69959 × 10^−13^	7.19949 × 10^−13^	5.78854 × 10^−13^

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
