# Peer review of "Evaluation of the Environmental Benefits Associated with the Addition of Olive Pomace in the Manufacture of Lightweight Aggregates"

_materials, 2020, doi:10.3390/ma13102351_

Round 1

Reviewer 1 Report

The paper entitled " Evaluation of the environmental benefits associated with the addition of olive pomace in the manufacture of lightweight aggregates" has great potential to be interesting for scientific society. 

Generally, the text of the paper is well written. The presentation is clear and technically correct. There are quite interesting research results.

However, the article structure presents some comings that must be addressed before publication.
I have comments:

The manuscript language and grammar should be carefully checked and edited.

Title

The title clearly describes the article.

Abstract

The abstract reflects the content of the article.

Introduction

Introduction is interesting, contains quite a lot of references to literature, and finally contains the specific purpose of the authors' research.

Methodology

The methods of research are solid and the materials are well documented.

Results

No statistical data (CV, SD) are given in Tables 2, 3. With this type of material which is aggregate, such data is important. whether there were differences, deviations from the average.

Line 342 – Is Tabale 2, should be Fig. 2.

The descriptions in Fig. 2, 3 are difficult to read. Authors should also place error bars on the charts. There are no statistics (SD).

The graphs show that the results between individual aggregates are negligible (statistically insignificant), but this should be demonstrated in a scientific article, support the testet.

I understand that the paper concerns LCA aggregate with olive pomace, but first of all it should be said how olive pomace affected the properties of the aggregate - its density, absorbability, strength, and only then Autohors can analyze the impact of waste on LCA. Then it makes sense. Do the authors have such results and can present them?

There isn’t discussion in this section.

It really needs to be revised and much improved, needs analysis of results in relation to literature. In its current form, the paper looks like a technical report (despite important and interesting research), there are no analysis in relation to existing research or scientific articles concerning the production of aggregate with similar waste.

This part of the article is important to present:

- the background and context

- related studies and actual knowledge

- why this study is pertinent and how could potentially improve the knowledge.

According my suggestion the paper needs major restructuration and complete discussion.
I recommend the paper to publish after major revisions.

Author Response

Reviewer #1: The manuscript language and grammar should be carefully checked and edited:

Thank you. We have corrected the English where necessary.

Title:

The title clearly describes the article.

Abstract:

The abstract reflects the content of the article.

Introduction:

Introduction is interesting, contains quite a lot of references to literature, and finally contains the specific purpose of the authors' research.

Methodology:

The methods of research are solid and the materials are well documented.

Thank you very much for your valuable comments.

Results:

No statistical data (CV, SD) are given in Tables 2, 3. With this type of material which is aggregate, such data is important. whether there were differences, deviations from the average.

The software used does not provide data of a statistical nature that can be displayed alongside the graphs. However, the database on which the measurements have been based, Ecoinvent v3.2, already incorporates the adjusted uncertainty in the data.

Line 342 – Is Table 2, should be Fig. 2.

In line 342, table 2 have been modified by fig.2.

The descriptions in Fig. 2, 3 are difficult to read. Authors should also place error bars on the charts. There are no statistics (SD).

The authors have been modified the descriptions in figures 2 and 3. The software used does not provide data of a statistical nature that can be displayed alongside the graphs.

The graphs show that the results between individual aggregates are negligible (statistically insignificant), but this should be demonstrated in a scientific article,

support the tested.

Some of the comparative results of the aggregates are shown in the reference article number 20. The low differences in the initial data generate results with small variability.

I understand that the paper concerns LCA aggregate with olive pomace, but first of all aggregate with olive pomace, but first of all it should be said how olive pomace affected the properties of the aggregate - its density, absorbability, strength, and only then Authors can analyze the impact of waste on LCA. Then it makes sense. Do the authors have such results and can present them?

The authors mention the results of the properties of the aggregates with olive pomace in a previous article cited in the text, line 102-112

Reference number [20] Moreno-Maroto, J.M., Uceda-Rodríguez, M., Cobo-Ceacero, C.J., Calero de Hoces, M. Martín-Lara, M.A., Cotes-Palomino, T., López García, A.B., Martínez-García, C., 2019. Recycling of ‘alperujo’ (olive pomace) as a key component in the sintering of lightweight aggregates. J. Cleaner Prod. 239, 118041. https://doi.org/10.1016/j.jclepro.2019.118041

There isn’t discussion in this section. It really needs to be revised and much improved, needs analysis of results in relation to literature. In its current form, the paper looks like a technical report (despite important and interesting research), there are no analysis in relation to existing research or scientific articles concerning the production of aggregate with similar

waste.

The authors have introduced the discussion in section 3.3 to improve it. Recent information on similar scientific papers has been incorporated in conclusions.

Reviewer 2 Report

This article presents an evaluation of the environmental benefits associated with the addition of olive pomace (OP) in the manufacture of lightweight aggregates (LWAs). Overall, the article conducted a life cycle assessment (LCA) to quantify the environmental benefits associated with the manufacture of LWAs based on clay and the incorporation of OP. The following comments are suggested:

  1. This article only discusses the different amounts of OP added from the economic perspective, and does not specify the properties of the LWAs produced. The author should clarify this point.
  2. In Table 1 and Table 2, some abbreviated words should be defined.
  3. Line 355, section 3.2 should be section 3.3.
  4. It is recommended that the author use the graphic method to present the steps of the life cycle assessment.
  5. It is recommended that the author present the results of life cycle assessment in the form of environmental performance indicators.
  6. The computer software used in this study was SimaPro version 8.3.0.0 from PRé Consultants. It is recommended that the author briefly introduce the operating procedures of the software.
  7. Lines 369-370, “The research conducted by Heijungs et al. (2007) [36] is noteworthy, in which they describe the phenomenon and its reasons.” The author should extract these important contents in this article.
  8. Lines 377-380, “An optimal amount could be between 1.25 and 2.5 wt% OP, even up to 3 wt% OP. These proportions would not negatively affect the technological properties of the resulting aggregates, but rather may improve them, as described by Moreno-Maroto et al. (2019).” The author should specifically point out those improvements.

Author Response

Reviewer #2:

This article only discusses the different amounts of OP added from the economic perspective, and does not specify the properties of the LWAs produced. The author should clarify this point.

The authors mention the results of the properties of the aggregates with olive pomace in a previous article cited in the text, line 102-112

Reference number [20] Moreno-Maroto, J.M., Uceda-Rodríguez, M., Cobo-Ceacero, C.J., Calero de Hoces, M. Martín-Lara, M.A., Cotes-Palomino, T., López García, A.B., Martínez-García, C., 2019. Recycling of ‘alperujo’ (olive pomace) as a key component in the sintering of lightweight aggregates. J. Cleaner Prod. 239, 118041. https://doi.org/10.1016/j.jclepro.2019.118041

In Table 1 and Table 2, some abbreviated words should be defined

The authors have defined the abbreviated in Table 1 and Table 2

Line 355, section 3.2 should be section 3.3.

In line 355, section 3.2 have been modified by section 3.3

It is recommended that the author use the graphic method to present the steps of the life cycle assessment.

The authors have introduced the new figure 1 with the methodology of the life cycle assessment.

It is recommended that the author present the results of life cycle assessment in the form of environmental performance indicators.

This possibility was considered, but due to the high number of samples and results, no graphs were obtained to clearly show the results. The authors chose to show the results according to the selected impact categories.

The computer software used in this study was SimaPro version 8.3.0.0 from PRé Consultants. It is recommended that the author briefly introduce the operating procedures of the software

The authors have introduced the new figure 2 with the methodology of operation of the software.

Lines 369-370, “The research conducted by Heijungs et al. (2007) [36] is noteworthy, in which they describe the phenomenon and its reasons.” The author should extract these important contents in this article.

The research conducted by Heijungs et al. (2007) showed that the normalization results for marine aquatic ecotoxicity might be biased, because this impact category indeed is not widely recognized, connected to a lot of substances, and not dominated by just a few substances.

Lines 377-380, “An optimal amount could be between 1.25 and 2.5 wt% OP, even up to 3 wt% OP. These proportions would not negatively affect the technological properties of the resulting aggregates, but rather may improve them, as described by Moreno-Maroto et al. (2019).” The author should specifically point out those improvements.

The research show that the addition of OP in low proportions (mainly 2.5%) is more positive to achieve aggregate bloating, pore formation and a lighter structure. The high calorific value of OP also helps to reduce the firing temperature significantly, which would result in energy savings.

Round 2

Reviewer 1 Report

The authors included most of my comments. I accept paper for publication.